# Morphoanatomy and Histochemistry of Septal Nectaries Related to Female Fertility in Banana Plants of the ‘Cavendish’ Subgroup

**DOI:** 10.3390/plants11091177

**Published:** 2022-04-27

**Authors:** Manassés dos Santos Silva, Adriele Nascimento Santana, Janay Almeida dos Santos-Serejo, Claudia Fortes Ferreira, Edson Perito Amorim

**Affiliations:** 1Posgraduate Program in Biotechnology, State University of Feira de Santana, Feira de Santana 44036-900, Bahia, Brazil; manasses.tec@hotmail.com; 2Department of Agricultural Sciences, Federal University of Recôncavo da Bahia, Cruz das Almas 44380-000, Bahia, Brazil; adriele.santanna@hotmail.com; 3Embrapa Cassava and Fruit, Cruz das Almas 44380-000, Bahia, Brazil; janay.serejo@embrapa.br (J.A.d.S.-S.); claudia.ferreira@embrapa.br (C.F.F.)

**Keywords:** banana, female fertility, inflorescence, reproductive barriers

## Abstract

The objective of this study was to gain a deeper understanding of the morphoanatomical and histochemical structures that compose the nectary of pistillate flowers (female), which are involved in the female fertility of banana plants belonging to the ‘Cavendish’ subgroup. The diploid Calcutta 4 and the Grand Naine cultivar were used for the assessment. Five stages of floral development were proposed. Pistillate flower nectaries were subjected to morphological characterization, morphoanatomy, and histochemical tests (phenolic compounds, proteins, and lipids). Morphoanatomical analysis revealed a greater presence of narrow nectariferous ducts and more developed pluristratified papillae in Calcutta 4. In contrast, Grand Naine displayed cell disintegration in nectariferous ducts and pluristratified papillae, absent transmitting tissue, and greater amounts of vascular bundles at anthesis. However, Calcutta 4 displayed no changes in the nectariferous duct at any of the stages. An association was found between phenolic compounds and lipids in vacuoles adjacent to the vascular bundles, with greater amounts found in Grand Naine. The localization of phenolic compounds may suggest that these compounds play a role in nectar secretion or the oxidation of the nectary region, ultimately limiting the growth and passage of the pollen tube and preventing ovule fertilization.

## 1. Introduction

Banana produces one of the most consumed tropical fruit in the world. As a result, banana trees are cultivated in extensive areas in the tropics and subtropics [1]. In 2019, the global production of bananas was estimated to be 116.8 million tonnes on 5.1 million hectares of land. In the same period, Brazil produced approximately 6.8 million tons on 461 thousand hectares and is the fourth main producing country in the world (after India, China, and Indonesia) [2].

Triploid banana varieties are the most commercialized variety on the world market, owing to their favorable agronomic characteristics, such as nutritional quality and palatability. However, the diploid genotypes are important for breeding, as they are resistant to pests and diseases and are tolerant to abiotic factors, such as water deficit and extreme temperatures [3,4]. Biotic factors limiting banana production include black Sigatoka (*Mycosphaerella fijiensis* Morelet (anamorph: *Pseudocercospora fijiensis* (Morelet) Deighton) and *Fusarium* wilt (*Fusarium oxysporum* f. sp. *cubense*) [5,6,7].

*Fusarium* wilt is one of the main biotic stress factors affecting banana plantations and one of the most destructive diseases, with emphasis on the tropical race 4 variant in cultivars of the ‘Cavendish’ subgroup, characterized by yellowing of young leaves, splitting of the pseudostem and, eventually, plant death [5,7,8,9,10,11]. Forecasts for the next 20 years estimate that tropical race 4 may affect 17% of commercial plantations worldwide, with losses of 36 million tons, equivalent to USD 10 billion [12].

Banana production is also affected by black Sigatoka, characterized by streaks that lead to leaf necrosis at more advanced stages, in addition to interfering with photosynthesis and reducing the quantity and quality of fruits [13,14]. Agrochemicals that control the disease have high costs, with annual expenditures of USD 1000/ha in large plantations, corresponding to up to 30% of total production costs [6,14].

As the chemical control of black Sigatoka increases cost production and may impact human health and the environment, and control has not been generated for *Fusarium* wilt, the development of resistant cultivars through hybridization is fundamental for mitigating the effects of these diseases on farms [13,15].

One of the most serious limitations to the genetic improvement of bananas is the low fertility of some commercial cultivars, which restricts hybridization to the formation of few seeds and small progenies or causes the complete absence of seeds. Although the ‘Cavendish’ subgroup has very low fertility, the ability of this cultivar to obtain seed is not limited, as already reported by the Fundación Hondureña de Investigación Agrícola (Fhia) [3,15,16,17,18].

Female sterility in banana plants is caused by several complex factors, which may act alone or together, such as the low germination rate of pollen grains, the slow and irregular growth of the pollen tube and the non-fertilization of ovules, and inhibition or failure of embryo sac development in pistillate flowers [16,19,20,21,22].

Information regarding the reproductive barriers in pistillate flowers that limit genetic improvement and induce fertility issues in banana plants is scarce. Some studies have revealed that the pistillate flower nectary of the Grand Naine cultivar undergoes oxidation/necrosis that prevents female fertility [23,24,25,26].

Understanding the structure and characterizing the chemical compounds in the septal nectary of pistillate flowers, as well as their relationship with sterility, may enable a better understanding of female fertility, in addition to providing subsidies for the development of new cultivars. Accordingly, the causes and the exact location of this barrier must be elucidated to enable the subsequent use of appropriate techniques to alleviate this barrier.

Considering the lack of information regarding reproductive barriers in banana plants, the objective of the present study was to gain a deeper understanding of the morphoanatomical and histochemical structures that compose the septal nectary of pistillate flowers of cultivars belonging to the ‘Cavendish’ subgroup.

## 2. Results and Discussion

Longitudinal sections of female banana flowers at different development stages and exposure to the nectary revealed that the Grand Naine cultivar had a region of oxidation in the nectary at S5, which was not observed in Calcutta 4 at the same floral development stage (Figure 1). Histochemical sections were analyzed, and both genotypes were compared, to investigate the morphology of this necrotic region and the chemical compounds that may be involved in Grand Naine nectary oxidation.

Septal nectaries located in the distal region of the ovary are formed by septa that together form a cavity. The location of the septal nectary in the distal region of the ovary in *Musa* spp. Was reported previously in *M. textilis* Nee [27], *M. errans* Teodoro var. botoan Teodoro [28], and *M. acuminata* Colla cv. Dwarf ‘Cavendish’ [29,30], corroborating the results of the present study.

Calcutta 4 was found to have smaller pistillate flowers than Grand Naine, according to the length and diameter of the septal nectaries presented in Table 1. The length of Calcutta 4 nectaries was significantly different, with larger values obtained at S4 (6.86 mm) and S5 (7.05 mm). Similarly, smaller diameters were obtained at S4 and S5, with 2.07 and 2.17 mm, respectively, for Calcutta 4. At S5, the Grand Naine cultivar had higher length and diameter values of 7.31 and 3.60 mm, respectively (Table 1).

These findings indicate that S1 and S2 were significantly different between the genotypes in terms of the length of the septal nectaries in the Grand Naine cultivar. However, only S5 had a significant difference in diameter in this cultivar. S5 corresponds to the time when the inflorescence is in a pendulous position on the pseudostem, with flowers open and ready to be pollinated (anthesis).

According to Waniale et al. [18], the length of the pistillate flower influences banana plant fertility. According to these researchers, as Calcutta 4 has smaller flowers, its seeds are formed along the entire length of the ovaries (fruits) because the pollen tube runs through the entire ovarian region. When pollinated with Calcutta 4 at anthesis, the diploids Mshale and Nshonowa (subgroup Mchare—AA), and the Enzirabahima cultivar (subgroup Matooke—AAA), which have larger flowers and low female fertility, partially displayed seeds in their fruits. Using the same approach, it is hypothesized that pistillate flowers pollinated at a younger stage of floral development promote increased pollen tube growth and seed formation in banana plants owing to the shorter path from the pollen tube to the ovules.

Morphoanatomical sections of the floral developmental stages of the septal nectaries of pistillate flowers are presented in Figure 2 and Figure 3 for Calcutta 4 and Grand Naine.

A cross-section of the septal nectaries displayed the presence of tubular canals or nectariferous ducts with ramifications, forming a complex and highly proliferated labyrinthine structure in both genotypes analyzed. Variations were observed in the nectariferous ducts, which formed a radiate pattern in a three-to-four-arm shape in Calcutta 4 (Figure 2B) and Grand Naine (Figure 3B,H). However, in Calcutta 4, because the flowers were smaller, the channels were narrower than those in Grand Naine. This narrowing in Calcutta 4 may favor the transmittal of the pollen tube, suggesting that the smaller the width of these canals, the greater the potential for fertility in the banana. Notably, narrowing may enable better orientation and passage of the pollen tube to fertilize the ovules.

Dense cells were formed near the nectar ducts, which constitute the nectariferous tissue in Calcutta 4 (Figure 2C,G,H) and Grand Naine (Figure 3C,G). These dense cells occurred in the locular region of the central axis of the pistillate flower (Figure 2A and Figure 3A). According to Schmid [31], in monocots, nectary canals constitute a cavity for sugar or nectar and occur in the septal region between adjacent carpels due to incomplete fusion of these carpels during the development of the gynoecium.

The epithelium that constitutes the nectariferous duct in the genotypes was formed by columnar cells and numerous pluristratified papillae, which lined the parenchyma (Figure 2D and Figure 3D). Calcutta 4 presented papillae formed by more elongated columnar cells (Figure 2D). Although the papillae of Grand Naine had cells with a columnar aspect, they were more agglomerated (Figure 3D).

These observations suggest that further studies should be carried out to identify possible substances released from multi-stratified papillae and that they are related to the transmittal of the pollen tube in banana trees. In Calcutta 4, these papillae possibly favor female fertility, as they are larger than those in Grand Naine, which appear to be damaged. Accordingly, Calcutta 4 facilitates the orientation and passage of the tube and the consequent fertilization of the ovules.

Fundamental parenchymal tissue was found to line the septal nectary in Calcutta 4 and Grand Naine, with a large number of parenchymal cells observed (Figure 2A and Figure 3A). However, in Grand Naine, this tissue had a greater spacing formed by the distance of the cells (Figure 3E,F). In Calcutta 4, the tissue was more compact, with occasional spaces between its cells (Figure 2E,F).

The fundamental or filling parenchyma is characterized by cells with thin primary walls, large vacuoles, and characteristic intercellular spaces [30], which were observed in greater quantity in the Grand Naine cultivar in the present study.

According to Ren and Wang [32], the parenchyma adjacent to the septal nectary usually exhibits two-to-six layers of small, thin-walled secretory cells in banana plants. In this study, the genotypes analyzed showed variations in the range of 3–7 layers of small cells adjacent to the nectary tissue (Figure 2C and Figure 3C), which does not allow an association with fertility due to the variation in layers between the genotypes and stages analyzed.

The size of the intercellular spaces observed in Calcutta 4 and Grand Naine increased toward the vascular zone, where the vascular bundles are located (Figure 2E and Figure 3E). These bundles consist of xylem and phloem, as they are located close to the septal nectary and are cylindrical in shape in Calcutta 4 (Figure 2E) and Grand Naine (Figure 3E). The number of vascular bundles was similar in all stages of floral development in the analyzed genotypes. Furthermore, the vascular bundles presented normal xylem and phloem orientation. However, the more external bundles were larger, while the more internal bundles were characterized as small irregular cylinders, located adjacent to the nectary, and might be amphicrival when the phloem involves the xylem (Figure 2E and Figure 3E).

The presence of numerous vascular bundles near the septal nectaries may increase nectar production, which is secreted directly from the epidermis of the carpel external wall and contributes to the vascularization of the nectary [30,33].

Substances, such as amino acids, proteins, mucilage, lipids, alkaloids, phenolic compounds (PCs), terpenoids, glycosides, organic acids, mineral ions, vitamins, antibiotics, and antioxidants, are reported to be an integral part of the nectar composition of different species [26,34,35].

The results obtained in the present study corroborate those of Kirchoff [33], who analyzed the ovary anatomy of three species of the Musaceae family (*M. velutina, M. ornata,* and *M*. cv. Go Sai Yung) and reported a similar septal nectary anatomical structure in the investigated species, such as the presence of epithelium in the duct formed by columnar and papillary cells.

Similar results were reported by Ren and Wang [32], who described the septal nectary of the species *M. basjoo, Ensete glaucum*, and *Musella lasicarpa*. These researchers observed epithelial cells, vascular tissue, and nectary opening similarities. However, the differences involved the shape of the longitudinal and cross-sectional sections, the arrangement of secretory cuticles, and the presence of fibrillar material.

The same nectary morphology was found in male and female flowers of *M. acuminata* cv. Dwarf ‘Cavendish’ by Fahn and Benouaiche [34]. According to these researchers, the secretion of the septal nectary occurs as a function of the organelles present in the fundamental parenchyma, which is coated by a uniseriate epidermis with numerous pluristratified papillae, thereby corroborating the findings of the present study (Figure 2D and Figure 3D).

Herein, transmitting tissue could only be observed in Calcutta 4 and was parallel to the nectary region (Figure 2J). However, this tissue was absent in Grand Naine (Figure 3J) but may occur anterior to the nectary region. The transmitting tissue is located just below the stigma and is centrally positioned along the entire length of the style to the ovary, presenting elongated cells with thick walls. Smith et al. [36] also observed similar characteristics of the transmitter tissue of Nicotiana tabacum, corroborating our study.

According to some authors, the transmitting tissue nourishes the pollen tube, as it is composed of, among other substances, polysaccharides and proteins, which are responsible for pollen growth [36,37,38]. Thus, there is a possible relationship between fertility in Calcutta 4 and the presence of the transmitting tissue as an indicator of pollen orientation, resulting in ovule fertilization. As mentioned, transmitting tissue was not observed in Grand Naine, which did not display pollen tube growth, thereby prohibiting fertilization.

Barbosa [25] used fluorescence microscopy to identify pollen tube growth in crosses involving Calcutta 4 and Grand Naine and Prata-Anã cultivars at anthesis, which corresponds to S5. A lower quantity of pollen tubes was observed in the lower third of the stipe in Grand Naine compared with Calcutta 4; this reduced number of pollen tubes was more significant in the nectary region, demonstrating that there are no physical barriers preventing fertilization in the diploid genotype analyzed. On the other hand, both triploid cultivars (Prata-Anã and Grand Naine) did not display penetration of pollen tubes, confirming the existence of some level of reproductive barrier that compromises fertilization.

The transmitting tissue divides into three channels toward each ovary loculus near the base of the stipe [33]. In this study, loculi were observed in Calcutta 4 and Grand Naine at all stages of floral development (Figure 2I and Figure 3I). The loculi correspond to the region where two-to-four rows of ovules per loculus are immersed in a mucilage, just below the septal nectary (distal region of the ovary) [16,39,40].

Grand Naine exhibited cell disintegration in septal nectary ducts at S5 (Figure 3J). Such disintegration was not observed in Calcutta 4, which had no nectary duct changes at any of the floral development stages (Figure 2H,J). S5 corresponds to the time when the inflorescence is in a pendulous position on the pseudostem, with flowers open and ready to be pollinated (anthesis).

Fahn and Benouaiche [34] reported that cultivars of the ‘Cavendish’ subgroup do not secrete nectar from the septal nectaries of pistillate flowers at some stages of floral development, and epithelium cells disintegrate even before anthesis, which corresponds to S4 in the present study.

According to Soares et al. [24], the Grand Naine cultivar shows signs of oxidation/necrosis in the septal nectaries of pistillate flowers at anthesis, which could act as a barrier to fertility, inhibiting pollen tube development and ultimately resulting in the formation of seedless fruits.

Barbosa [25] observed morphoanatomical differences in the septal nectary of pistillate flowers in crosses involving Calcutta 4 and Grand Naine and Prata-Anã cultivars at anthesis, which corresponds to S5. The cultivars exhibited oxidation/necrosis in the nectary region, which was not observed in Calcutta 4, thereby corroborating the findings of the present study. Thus, this disintegration process results in a barrier capable of interrupting the growth of the pollen tube, consequently preventing ovule fertilization. The production of chemical compounds may also hinder or inhibit fertilization.

As shown in Figure 4, the histochemical tests revealed the presence of PC, total proteins, and lipids in the developmental stages of the septal nectaries in Calcutta 4 and Grand Naine.

Total PCs were evident in fundamental parenchyma cells near the vascular bundles and adjacent to the nectary in Calcutta 4 and Grand Naine (Figure 4A–C,J–L). However, more PCs, which were apparently contained in the vacuoles of the septal nectaries, were dispersed in parenchyma cells at developmental stages S3 (Figure 4K) and S5 (Figure 4L) of the Grand Naine cultivar, compared with the same stages in Calcutta 4. This finding may be related to the oxidation that occurs in the nectary at anthesis, corresponding to S5, as mentioned by Soares et al. [24], which promotes a barrier capable of preventing the growth of the pollen tube.

This microscopic observation of visualization made it possible to verify the presence of PC in the nectaries, which corroborates the study carried out by Silva et al. [26], who evaluated the presence of these compounds and that are involved in female fertilization in banana plants (diploid Calcutta 4 and cultivar Grande Naine) at different stages of floral development. The authors observed that the E3 stage presented higher significant values, with a decrease as the E5 stage was reached. Comparing Calcutta 4 and Grande Naine, amounts of phenolic compounds were obtained with values of 32.43 and 36.18 mg GAE.g^−1^, respectively. This study showed that in the septal nectaries of the pistillate flowers, phenolic compounds can be indicators of female fertility since they show similar effects in the diploid Calcutta 4 when compared with the cultivar Grand Naine.

The location of PC adjacent to the septal nectaries may suggest that it plays a role in nectar secretion or the oxidation of the nectary region, limiting the growth and passage of the pollen tube and preventing ovule fertilization in cultivars of the ‘Cavendish’ subgroup. This last supposition may be reinforced by the lower levels of these compounds in septal nectaries of fertile diploids, as mentioned also by Silva et al. [26].

According to Rocha [41], total PCs in the distal portion of the ovary (septal nectary) of pistillate flowers of two diploids improved with AA genome (089087-01 [(Malaccensis × Sinwobogi) × (Calcutta 4 × Heva)] and TH 0301), and the Prata-Anã (AAB) and Grand Naine (AAA) cultivars at anthesis, corresponding to S5. High total PC concentrations were observed in triploid genotypes, while low concentrations were found in the diploids.

PCs are secondary metabolites associated with different biological activities in plants [42]. Several factors may influence the quantity and distribution of PCs, mainly the species/cultivar [43,44]. The accumulation of these compounds in the cell leads to the intensification of free radical production, resulting in greater oxidation of plant tissues [42].

The test used to verify total proteins (TP) revealed the presence of this chemical component with reddish coloration in the cytoplasm of the nectary cells, and a greater amount in the pluristratified papillae of Calcutta 4 (Figure 4D–F) and Grand Naine (Figure 4M–O). These papillae may be related to the release of signaling substances for pollen tube orientation, thereby influencing ovule fertilization in banana plants.

Proteins are naturally present in plant cells, and their activities are related to the amount of substrates present [45]. Based on the findings of this study, the amount of PC identified may be a substrate for oxidative enzymes, such as peroxidase and polyphenol oxidase, enabling a better understanding of the occurrence and relationship of oxidation in plant tissues and cell disintegration at the S5 stage, as observed in cultivars of the ‘Cavendish’ subgroup.

Silva et al. [26] evaluated the presence of total proteins and enzymatic activities (peroxidase and polyphenoloxidase) in pistillate flowers of diploid Calcutta 4 and cultivar Grand Naine at different stages of floral development. The E3 stage also showed higher significant values. The analyzed genotypes (Calcutta 4 and Grand Naine) showed the following values, respectively: Total Proteins (3.20 and 2.42 µMoles min^−1^ g^−1^, peroxidase (0.89 and 0.78 µMoles min^−1^ g^−1^), and polyphenoloxidase (3.75 and 2.76 µMoles min^−1^ g^−1^). This study showed that enzymatic activities can be indicators of female fertility, suggesting the performance of crosses from the E3 stage, in cultivars of the ‘Cavendish’ subgroup, to verify the formation of fruits with seeds. The test for lipids revealed that these biomolecules are located in cells containing the total PC and adjacent to the vascular bundles in Calcutta 4 (Figure 4G–I), with a greater amount in Grand Naine at S4 (Figure 4P–R).

In addition to serving as biomolecules of energy reserve for the cell, lipids contribute to hydration that will enable the growth of the pollen tube through the transmitting tissue of the stipe, besides being a nectar constituent [34,46,47]. In the present study, the transmitting tissue was not found in the Grand Naine cultivar (Figure 3J). Thus, it is hypothesized that, for the pollen tube to perform its orientation journey through the stipe and fertilize the ovules, hydration of the transmitting tissue is necessary for the growth of the tube and exchange of signals between the pollen tube and the ovule.

Based on our results, there is variability in PC between diploids and triploids. The level of PC is associated with female fertility in banana plants. As Grand Naine, whose degree of sterility is high, has a greater amount of PC, fertilization could be hindered by this high level. Further, the diploid Calcutta 4, which is fertile, was found to have a low amount of these compounds.

Comparative studies between septal nectaries of *Musa* spp. species/cultivars are very important, especially to understand the relationship between pollen tube growth and female fertilization in triploids. Nectary structure and developmental patterns may provide a potential solution to fertilization limitations in biochemical and/or physical processes. Thus, the knowledge acquired in the present study will serve as a basis for the development of strategies to significantly improve the crop.

Studies involving pollination, fertilization, compatibility, seed production, regeneration, and dormancy are needed to select efficient parental combinations to develop hybrids and assist breeding programs [18,48].

Female fertility in banana plants has several meanings, depending on the scale adopted and whether ovules, pollen, or seeds are considered. For broader banana breeding studies, fertility is often expressed as the number of seeds per bunch (i.e., the greater the number of seeds, the higher the fertility) [48].

The absence of seeds and, consequently, sterility in banana plants may result from intense agronomic selection and a reflection of the domestication process of the species over the years. According to some researchers, chromosomal translocations and abnormal meiosis, such as embryo sac abortion, are causes of infertility that have no direct genetic origin but are instead dependent on specific conditions [3,20,21]. Furthermore, the A genome of *Musa acuminata* (AA) is assumed to contribute to female infertility in ancestral banana genomes, leading to parthenocarpy. However, due to its inherent properties, such as male and/or female sterility and unreduced gamete formation, genetic factors conditioning parthenocarpy have not been fully elucidated [49,50].

## 3. Materials and Methods

### 3.1. Plant Material

Two banana genotypes were selected according to the degree of female fertility: diploid Calcutta 4 (considered fertile) and the Grand Naine cultivar of the ‘Cavendish’ subgroup (considered to have a high degree of sterility) [16,17,24,48] were employed. The seedlings were obtained from the Banana and Plantain Germplasm Bank of Embrapa Cassava and Fruits, in Cruz das Almas, Bahia, Brazil (12°39′13″ south latitude and 39°07′21″ west longitude of Greenwich) [51] and planted in May 2017. The plants were maintained in the field and managed according to the recommendations for this crop [52]. For the analyses, 25 plants were used for each genotype.

### 3.2. Flower Development Stages

Five stages of floral development were proposed for the genotypes used in the study (Figure 5).

This proposal was based on the classification established by Fortescue and Turner [21], which refers to the inflorescence position in banana plants and includes the vertical position (apex pointing upward), horizontal position to the ground, and downward position (apex toward the ground). These stages were used to collect inflorescences for morphoanatomical and histochemical analysis. Five plants per stage were used for each genotype (Calcutta 4 and Grand Naine).

### 3.3. Morphoanatomical and Histochemical Characterization

Pistillate flowers were collected from the bunches to remove the septal floral nectaries and carry out subsequent morphological characterization. Fresh samples were employed to measure the length and diameter of the septal nectary; photographs were obtained using a stereoscopic microscope (Z4W model, Leica^®^, Wetzlar, Germany), and measurements were determined using the ImageJ software. All analyses were performed using three nectaries per stage, with three repetitions (replicates) for each genotype.

For the anatomical analysis, the septal nectaries were fixed in FAA70 solution (formaldehyde, glacial acetic acid, and alcohol) [53], subjected to vacuum for 24 h, and then transferred to ethyl alcohol (70%). The material was placed in blocks using Leica^®^ Historresin [54] for solidification, according to the manufacturer’s recommendations. Subsequently, longitudinal and cross-sectional sections were obtained on a rotating microtome (1516 model, Wild Leitz, Heerbrugg, Switzerland) at a thickness of 8 µm. The slides were stained with toluidine blue at 0.05% in acetate buffer, pH 4.3 [55], and mounted between the slide and coverslip with synthetic resin [56].

In order to analyze the main groups of secondary metabolites, subsequent longitudinal and transverse sections were performed on a rotating microtome (model 1516, Wild Leitz, Heerbrugg, Switzerland), with a thickness of 8 µm, and subjected to the following histochemical tests: 10% ferric chloride [53] to detect total non-structural phenolic compounds (PC), Xylidine ponceau to detect total protein (TP) [57], and Sudan Black B [58] to detect lipids. Protein and lipids tests were carried out through double staining, according to the methodology of [59]. The analyses were performed using a light microscope (Bx51 model, Olympus, Tokyo, Japan). Three repetitions were used per slide, considering one nectary in each repetition.

## 4. Conclusions

In triploid banana cultivars (AAA), sterility hinders the genetic improvement of the crop. Therefore, it is very important to understand the morphoanatomy of the flower, the pollen–pistil interaction, and the dynamics of pollen tube growth, as only a few studies have been conducted on the sterility process. Furthermore, information on the structure and composition of the septal nectaries of banana flowers is scarce. Nonetheless, the hypotheses proposed in this study may help future research. The knowledge acquired will also serve as a basis for the formulation of strategies to overcome the fertility issue in banana plants, ultimately assisting with the development of new cultivars of the ‘Cavendish’ subgroup. The use of new techniques, such as fluorescence microscopy, is recommended to verify the dynamics of the pollen tube in the stages analyzed, in addition to histochemical tests to identify substances and the chemical composition of the nectar to verify its influence on pollen tube development and fertilization in cultivars of the ‘Cavendish’ subgroup.

## Figures and Tables

**Figure 1 plants-11-01177-f001:**
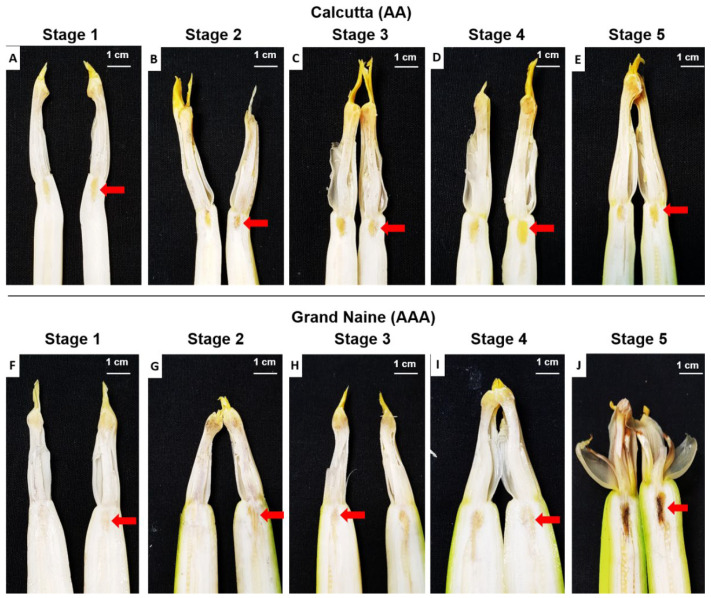
Longitudinal section of female flowers of Calcutta 4 and Grand Naine showing the nectary at different development stages. (**A**,**F**) stage 1: partial inflorescence emission from the pseudostem – vertical position (inflorescence base not visible); (**B**,**G**) stage 2: total emission of the inflorescence from the pseudostem – vertical position (visible base of the inflorescence); (**C**,**H**) stage 3: total inflorescence emission – horizontal position in relation to the ground; (**D**,**I**) stage 4: inflorescence pendulous to the pseudostem with closed flowers (pre-anthesis); (**E**,**J**) stage 5: Inflorescence pendulous to pseudostem with open flowers (anthesis) where Grand Naine displays nectary oxidation, while Calcutta 4 does not. Red arrow: Septal nectary.

**Figure 2 plants-11-01177-f002:**
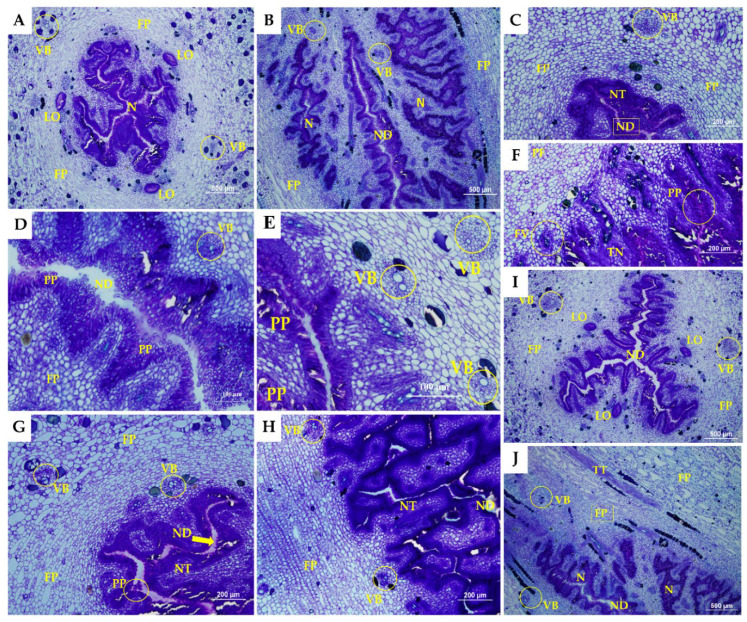
Section of developmental stages of the septal nectaries of Calcutta 4. Longitudinal sections: (**A**,**C**,**E**,**G**,**I**). Cross-sectional sections: (**B**,**D**,**F**,**H**,**J**). ND, nectariferous duct; VB, vascular bundle; LO, loculus; N, nectary; FP, fundamental parenchyma; PP, pluristratified papillae; NT, nectariferous tissue; TT, transmitting tissue. The white bars represent 100 µm (**D**,**E**), 200 µm (**C**,**F**–**H**), and 500 µm (**A**,**B**,**I**,**J**).

**Figure 3 plants-11-01177-f003:**
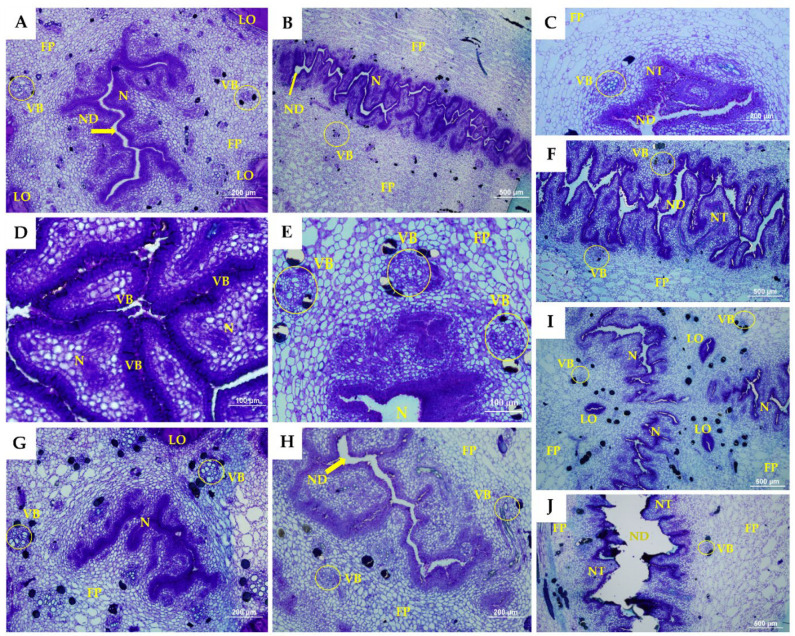
Section of developmental stages of the septal nectaries of Grand Naine. Longitudinal sections: (**A**,**C**,**E**,**G**,**I**). Cross-sectional sections: (**B**,**D**,**F**,**H**,**J**). ND, nectariferous duct; VB, vascular bundle; LO, loculus; N, nectary; FP, fundamental parenchyma; PP, pluristratified papillae; NT, nectariferous tissue. The white bars represent 100 µm (**D**,**E**), 200 µm (**A**,**C**,**G**,**H**) and 500 µm (**B**,**F**,**I**,**J**).

**Figure 4 plants-11-01177-f004:**
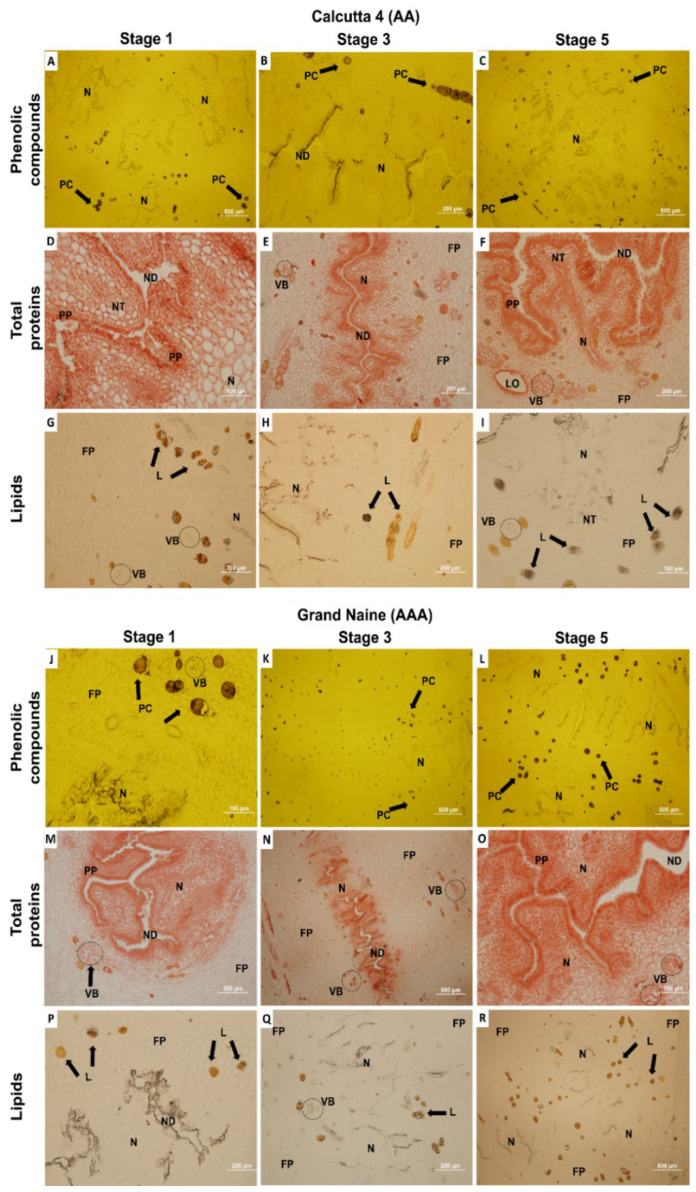
Sections representing the developmental stages of septal nectaries of Calcutta 4 and Grand Naine subjected to different histochemical tests. Longitudinal sections: (**A**,**C**,**D**,**F**,**G**,**I**,**J**,**L**,**M**,**O**,**P**,**R**). Cross-sectional sections: (**B**,**E**,**H**,**K**,**N**,**Q**). PC, phenolic compounds; ND, nectariferous duct; VB, vascular bundle; L, lipids; LO, loculus; N, nectary; FP, fundamental parenchyma; PP, pluristratified papillae. The white bars represent 100 µm (**D**,**I**,**J**,**O**), 200 µm (**B**,**E**,**F**,**G**,**H**,**M**,**P**,**Q**), and 500 µm (**A**,**C**,**K**,**L**,**N**,**R**).

**Figure 5 plants-11-01177-f005:**
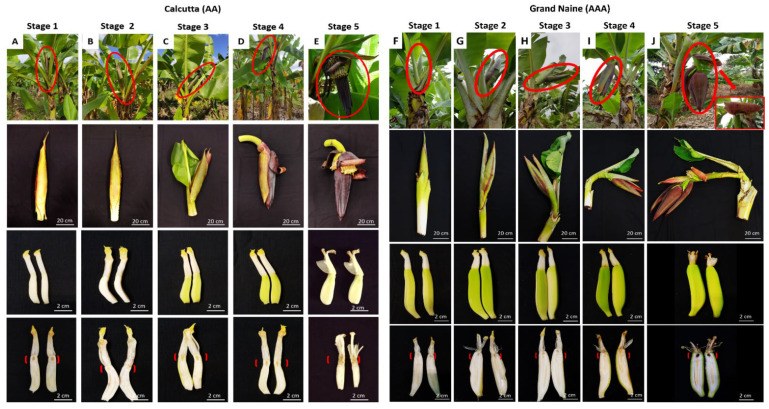
Proposed developmental stages of pistillate (female) flowers of the diploid Calcutta 4 and Grand Naine cultivars: (**A**,**F**) partial emission of the inflorescence from the pseudostem—vertical position (inflorescence base not visible); (**B**,**G**) total emission of the inflorescence from the pseudostem—vertical position (visible base of the inflorescence); (**C**,**H**) total emission of the inflorescence—horizontal position to the ground; (**D**,**I**) inflorescence in pendulous position to the pseudostem with closed flowers (pre-anthesis); (**E**,**J**) inflorescence in pendulous position to the pseudostem with open flowers (anthesis). [ ], Distal region of the ovary (septal nectary).

**Table 1 plants-11-01177-t001:** Morphometric characteristics of septal nectary length and diameter of Calcutta 4 and Grand Naine cultivar at the five stages of floral development.

Stages	Length (mm)		Diameter (mm)	
	Calcutta 4	Grand Naine	Calcutta 4	Grand Naine
S1	3.85cB	5.49bA	0.93cA	1.87cA
S2	4.44cB	5.21cA	1.68bA	2.87cA
S3	5.27bA	5.37cA	1.70bA	2.20cA
S4	6.86aA	6.29bA	2.07aA	2.68bA
S5	7.35aA	7.31aA	2.17aB	3.60aA
CV (%)	20.49		33.61	

Means followed by the same letter do not differ statistically, with lower case in the column and upper case in the row, based on the Scott Knott test at 5% probability. CV: coefficient of variation.

## Data Availability

All data were found in Embrapa Mandioca e Fruticultura, Brazil.

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
