# Peer review of "Morphoanatomy and Histochemistry of Septal Nectaries Related to Female Fertility in Banana Plants of the ‘Cavendish’ Subgroup"

_plants, 2022, doi:10.3390/plants11091177_

Round 1
Reviewer 1 Report
Very interesting investigation which is written very well and combining morphological analysis with microscopy of chemical main constituents such as polyphenols, lipids and protein in order to understand fertility in banana plants.
As illustrated in Figure 4, it looks like that there are more polyphenols in the Grand Naine at stage 5, which are supposed to be the cause for infertility of the plants. However, it is not really clear, how the authors of the study come up with the hypothesis, why the polyphenol content is higher. How was the total amount quantified? It is also not really clear which kind of polyphenols are detected with this technology – the condensed (oxidised) ones, the monomers or all together? There is a literature source cited for the methodology, however the article is in Portuguese and therefore not available/understandable for all. Microscopic techniques, localizing specific chemical constituents in plant tissues are surely a valuable tool for visualisation. However, these techniques not allow quantitative statements. At least it has to be better explained in the manuscript how the quantitative statement/hypothesis can be extracted out of it. It would be certainly very interesting to have a more detailed look on the molecular constitution of the septal nectaries with high resolution techniques such as HPLC-MS/MS (targeted or even untargeted metabolomics) and also combine them with the microscopic analysis. This can lead to a deeper understanding of the molecular mechanisms which influence fertility.
Further comments:
Message of paragraph line 388 – 394 should be transferred to the conclusion section. From my point of view it’s not very common to draw conclusions in the materials and methods part.
Author Response
Point 1: As illustrated in Figure 4, it looks like that there are more polyphenols in the Grand Naine at stage 5, which are supposed to be the cause for infertility of the plants. However, it is not really clear, how the authors of the study come up with the hypothesis, why the polyphenol content is higher. How was the total amount quantified? It is also not really clear which kind of polyphenols are detected with this technology – the condensed (oxidised) ones, the monomers or all together?
Response 1: Adjusted text. To explain the greater amount of phenolic compounds in stage 5, a bibliographic reference by Silva et al. (2021) who performed quantitative analyzes in Grand Naine and Calcutta 4.
Point 2: There is a literature source cited for the methodology, however the article is in Portuguese and therefore not available/understandable for all.
Response 2: The literature was replaced by another in English by the same authors.
Point 3: Microscopic techniques, localizing specific chemical constituents in plant tissues are surely a valuable tool for visualisation. However, these techniques not allow quantitative statements. At least it has to be better explained in the manuscript how the quantitative statement/hypothesis can be extracted out of it. It would be certainly very interesting to have a more detailed look on the molecular constitution of the septal nectaries with high resolution techniques such as HPLC-MS/MS (targeted or even untargeted metabolomics) and also combine them with the microscopic analysis. This can lead to a deeper understanding of the molecular mechanisms which influence fertility.
Response 3: We agree with the reviewer´s comment, however, our results should not be disregarded due to lack of the HPLC analysis. Our work aimed to present the scientific community new knowledge about banana fertility, a not very well understood, and limiting factor in bananas. It does not propose in any way to liquidate the topic, but to add information to the information that already exists and allow for the exploration for broadening new future perspectives, including HPLC analysis. We in the near future will be using HLPC in samplings of the same stages of floral development presented here in this manuscript. Furthermore, proteomic analysis of the latter samplings is undergoing as part of a thesis of a Doctoral fellowship to increase knowledge of the referred topic. Therefore, we hope the reviewer will take into consideration of the importance of the results presented in this manuscript regarding banana fertility, which certainly will contribute to the topic for bananas of the Cavendish subgroup.
Point 4: Message of paragraph line 388 – 394 should be transferred to the conclusion section. From my point of view it’s not very common to draw conclusions in the materials and methods part.
Response 4: Ajusted.
Reviewer 2 Report
This manuscript needs to be improved in order to be published in the journal.
Here are my suggestions:
At the level of figure 1- I would suggest indicating where the nectary is.
Table 2 is missing – it is referred to several times
Regarding figures 2, 3, and 4: - must be specified which is the magnifying power of microscope lenses? Were all preparations studied with the same objective? What are those or that?
From figures 2E and 3E, it is challenging to characterize the vascular bundles maybe if the images are enlarged.
Line 425 and 430- With what objectives the preparations were studied?
Line 426 - How were obtained the sections for histochemical tests?
Author Response
Point 1: At the level of figure 1- I would suggest indicating where the nectary is.
Response 1: Ajusted.
Point 2: Table 2 is missing – it is referred to several times
Response 2: Ajusted.
Point 3: Regarding figures 2, 3, and 4: - must be specified which is the magnifying power of microscope lenses? Were all preparations studied with the same objective? What are those or that?
Response 3: Ajusted.
Point 4: From figures 2E and 3E, it is challenging to characterize the vascular bundles maybe if the images are enlarged.
Response 4: Ajusted.
Point 5: Line 425 and 430- With what objectives the preparations were studied?
Response 5: Ajusted.
Point 6: Line 426 - How were obtained the sections for histochemical tests?
Response 6: Ajusted.

Round 2
Reviewer 2 Report
In my opinion the manuscript can be published in this form